# The Use of Virtual and Computational Technologies in the Psychomotor and Cognitive Development of Children with Down Syndrome: A Systematic Literature Review

**DOI:** 10.3390/ijerph19052955

**Published:** 2022-03-03

**Authors:** Elvio Boato, Geiziane Melo, Mário Filho, Eduardo Moresi, Carla Lourenço, Rosana Tristão

**Affiliations:** 1Department of Physical Education, Catholic University of Brasília, Brasilia 71966-700, Brazil; elvioboato@gmail.com (E.B.); geizianemelo93@a.ucb.br (G.M.); 2Center for Science and Technology-CogniAction Lab, Catholic University of Brasilia, Brasilia 71966-700, Brazil; braga@ucb.br (M.F.); moresi@ucb.br (E.M.); 3Department of Sport of Science, Universidade da Beira Interior, 3510-774 Covilhan, Portugal; 4Faculty of Medicine, University Hospital of University of Brasilia, Brasilia 70297-400, Brazil; rmtt@unb.br

**Keywords:** Down syndrome, virtual technologies, computational technologies, assistive technologies

## Abstract

Individuals with Down syndrome (DS) have numerous comorbidities due to trisomy 21. However, virtual reality-based therapy (VRT) has been used nowadays as a learning and visual motor tool in order to facilitate the development and learning process of this group. The aim of this article was to carry out an integrative review of the literature on the use of virtual and computational technologies in the stimulation of children with DS. A search was carried out according to the Preferred Reporting Items for Systematic Reviews and Meta-Analyses (PRISMA) through single key words or their combinations using AND or OR operators: “Down syndrome” AND (“development” OR “cognition” OR “visomotor” OR “digital game” OR “virtual reality”). Eventually, 18 articles were included in our review. The games used in the research were able to stimulate, through the visual field, global motor skills, balance, body scheme and spatial organization, in addition to the learning of mathematical concepts, in order to directly influence the autonomous life activities, language skills, social skills and educational aspects of people with DS. Electronic games contribute to the teaching-learning relationship and stimulate neuropsychomotor and cognitive functions and development in children with DS.

## 1. Introduction

Down syndrome (DS) is the most frequently diagnosed chromosomal disorder in newborns [1,2,3], formerly scientifically described by John Langdon Down in the second half of the 19th century as “Mongolian family” and then “Mongol idiot” [4] based on their innate physical characteristics. The cause of DS was only discovered in 1959 by Jerome Lejeune, when he realized that individuals with the syndrome had an extra chromosome in their karyotype, totally or partially accompanying chromosome 21. Thus, DS became understood as an imbalance in the chromosomal constitution of chromosome 21, and the incidence is one in 732 births, though the prevalence may have variability among racial/ethnic groups [5], with prevalence estimates ranging from 6.1 to 13.1 per 10,000 people [6]. There are physical features that characterize DS, the most common being the brachycephalic head, epicanthal folds, flat nasal bridge, eyelid fissures tilted upward, small mouth and ears, excessive skin on the nape, single transverse palmar furrow and fifth short finger, in addition to a deep plantar groove between the first and second fingers [3,7].

### Considerations about Comorbidities Associated with DS

DS affects different systems in the human body, with a negative impact on general development (see summary in Table 1). Among the health-related problems, people with DS may experience congenital heart disease in 40–60% of cases [3,8,9], muscle hypotonia [2], thyroid disorders [10], a behavioral phenotype [11], obesity and early aging [12]. One of the more frequent characteristics of DS is the generalized ligamentous hyper-laxity that occurs in about 43% of the cases [13]. As a result, people with DS may have atlantoaxial instability [14,15], which consists of increasing mobility between the first and the second cervical vertebrae (atlas and axes). Such instability can prevent or hinder certain movements that involve the cervical spine and neck muscles, and there must be careful monitoring regarding the possibilities of motor activities that the individual may or may not perform, remembering that, even with such atlantoaxial instability, the person must and can do physical activities, as long as the limitations related to the activity and its real possibilities are considered.

Congenital heart disease is considered to influence all developmental domains of children with DS, including cognition, expressive language [24,26,34] and gross motor function [19]. It is pointed out that DS is the most frequent cause of intellectual disability [14,35] which, in 95% of cases, is caused by the simple trisomy of chromosome 21 resulting from the non-disjunction of chromosomes in the meiotic division [36]. Abnormalities in the development of the cortex and cerebellum, in the first months of life, are probably substrates for posterior neurocognitive impairment [17]. Moreover, 75% of children with DS are at risk of hearing loss [7], and DS is also often associated with a wide range of eye complications, including refractive errors, eyelid abnormalities, strabismus, nystagmus, abnormalities in the lacrimal duct and iris, the presence of keratoconus and congenital or developmental cataracts [37]. Hearing and vision problems are associated with the fact that children with DS may exhibit functional brain connectivity being interrupted in the motor and prefrontal cortex [19], and these problems can mean that intellectual disability can be potentiated by disorders of sensory–perceptual processing, which can be neglected in the clinical segment of development, especially in the early years of life, when areas such as language and cognition are in the full phase of acquisition and consolidation. Motor issues are a consequence of these problems [21], making the search for alternatives that improve the neurodevelopment and psychomotor development of children with DS relevant, since the sum of all the factors linked to the trisomy of chromosome 21, and even considering each comorbidity separately, contribute significantly to delayed development. In summary, an estimate of 80 clinical conditions occur more frequently in people with DS than in those without the syndrome, but not all conditions manifest themselves in all individuals [27].

In view of the clinical conditions presented in DS, especially those that can interfere with the use of virtual and computational technologies, such as visual and hearing disorders, generalized hypotonia and ligament laxity, a recent review concluded that virtual reality-based therapy (VRT) combined with physiotherapy can improve motor proficiency in children and adolescents with DS [38]. Another study, investigating how gestural interaction with VRT improves the cognitive visual–motor abilities in adolescents with DS, showed a significant improvement in their cognitive visual–motor abilities [39].

VRT, also known as exergames, has become an important instrument that facilitates the development and learning process of children in the school environment, stimulating early and rehabilitation [22,32,40]. Thus, the objective of this article was to carry out an integrative review of the literature on the use of virtual and computational technologies in the stimulation of children with DS.

## 2. Literature Search

This review follows guidelines based on the Preferred Reporting Items for Systematic Reviews and Meta-Analyses (PRISMA) initiative (http://www.prisma-statement.org/) for bibliographic research and data communication in systematic reviews. The study was carried out through an integrative literature search, with the analysis of articles published between the years 2011 and 2020 in the PubMed, Scopus, Science Direct, EBSCO, SciELO and Web of Science databases. The following descriptors that composed the thematic filter were used together with the Booleans “And” and “Or”: “Down syndrome” AND (“toddler” OR “development” OR “children” OR “cognition” OR “visuomotor” OR “digital game” OR “virtual reality”).

Subsequently, as exclusion criteria, the following requirements were used: repeated articles, conference abstracts, research conducted with animals or human adults, monographs, dissertations and theses were excluded. Studies classified as literature review articles or meta-analytical studies and those that did not present in their summary the methodology and conclusions or results of the investigation were also excluded. The final number of articles selected was 18.

## 3. Data Collection

The citations gathered were independently extracted and examined for later comparison, and redundancies were eliminated. The reviewers analyzed the title and abstract of each article. The full texts of the selected manuscripts were read. The articles found were scrutinized for the study design, exclusion criteria, sample characterization, sample size, description of games or platforms, type of the intervention and main findings.

The initial search identified of 38.608 citations and, as this search returned a high number of results, we chose to apply a co-occurrence or bibliometric analysis of keywords to refine the search. In general, bibliometrics is the application of mathematical and statistical methods to books and other means of written communication [41], overring publications in general. A bibliometric network consists of graphs that comprise nodes (units of analysis) and edges (types of analysis). The nodes can be, for example, publications, journals, researchers, countries, organizations or keywords. Edges indicate relationships between pairs of nodes. The types of relationships most studied employ bibliometric methods, including those of citation, the co-occurrence of keywords and co-authorship. The co-occurrence of keywords or indexing terms, using network analysis, reveals the conceptual structure of a research field [42].

The co-occurrence analysis of keywords was used to identify trends in the focus of the topics studied in this research. Zupic and Cater [43] stated that when words often co-occur in documents, it means that the concepts of those words are closely related. The result of the co-occurrence analysis is a network of themes and their relationships that represents the conceptual space of a field. Co-occurrence analysis is based on keywords contained in the documents. For the bibliographic search, the following search expression was used: “Down syndrome” AND ((“development” OR “cognition” OR “visuomotor” OR “digital game” OR “virtual reality”) AND (“children” OR “toddler”). Here, 8508 documents were retrieved from the Scopus database and 6327 from Web of Science. The two sets of metadata were merged, and duplicate records were eliminated. The final set of metadata resulted in 13,950 documents.

The metadata was imported into VOSviewer [44] and the authors’ keyword co-occurrence network was generated [43]. Figure 1 shows the resulting network, which has 550 nodes, 5235 edges and 6 clusters, with a minimum of five times of co-occurrence. The keyword “Down syndrome”, which was the main search argument for bibliographic research, is the most relevant. Other keywords also stand out: intellectual disabilities, autism spectrum disorder, children, language, quality of life, etc.

To determine the most influential keywords, the network was exported in GML format and imported into Gephi [45], which is software for network analysis. Table 2 shows the most influential keywords in the co-occurrence network. The influence is defined by eigenvector centrality [46].

To identify the research fronts, a network of citation for references cited was generated at least 40 times. The resulting network has 245 nodes, 10,834 edges and 4 clusters, as shown in Figure 2. To determine the most influential documents, the graph was exported in GML and imported into Gephi [45]. After calculating the average degree and eigenvector centrality, the documents were ordered and the 10 most influential were selected, which are presented in Table 3.

Figure 3 presents the conceptual space of the research, with the selection of the keywords “Down syndrome” and “games”. The graph was obtained from reading the metadata of the bibliographic search result using the software VOSviewer [44]. The lists of terms and edges were exported and retrieved in an Excel spreadsheet to be analyzed in software yEd, which is free software for editing graphs. The communities were categorized based on the keywords with the highest frequency of occurrence. This type of representation is useful to identify the conceptual routes, which allows for further research.

After identification, 24,922 duplicate articles were removed. The remaining 13,689 articles were exclusively related to infants or children with DS. When performing the screening through the inclusion criteria, 67 studies were included with the descriptors: “Visuomotor Development” (22) OR “Digital Games” (10) OR “Virtual Reality” (35) (Figure 4). These publications were evaluated based on their titles and abstracts using the described inclusion criteria, and 47 papers were excluded after this. These final 20 studies were entirety read, and 18 were selected as a result, as two were excluded for conflicts with the indexation code of the journal or article. No studies relevant to this review were found among the references analyzed in the gray literature. These stages of the search and selection process are described in Figure 4. The following variables were collected for each selected article: authors and year of publication, country where the study was conducted, type of technology applied in the study and the main results. This information is shown in Table 4.

## 4. Results

To structure the findings of this study, the following sections represent the categories of main findings that are described as: the motor development of children with DS; the development of communication in children with DS in the context of infocommunication, i.e., combining cognitive informatics [39]; and the use of electronic games for the assessment and stimulation of children with DS. Table 4 presents the result of the research, including the main findings of the articles found.

### 4.1. Down Syndrome and Motor Development

Several studies on the development of children with DS point to the fact that their comorbidities lead to a delay in general development in psychomotor aspects, even considering that the development follows the same characteristics of people without the syndrome [20,21,58,63,64,65,66]. One of the main problems relates to static and dynamic balance and the spatial-temporal gait parameters that are statistically and significantly different between children with and without DS, also considering that the balance and walking ability of typically developing children improves during growth, while in children with DS they remain low despite independent walking [63].

Another problem is related to the temporal organization and body scheme, which are far below expectations, even though fine motor development presents little damage in most cases [65]. Regarding the development of spatial organization in DS, a study with the use of external clues (such as maps), found a low performance in relation to the recognition and sequential organization of landmarks and references [64]. In addition to these considerations, there are motor changes in children with DS who exhibit a delay in the acquisition of motor skills compared to children without the syndrome, a fact that can impair tasks that require manual dexterity and grip strength [67].

Thus, it can be inferred that there are delays in all aspects of the development of the body schema of DS children that are due to perceptual and motor deficits, which are responsible for the formation of the same and functional activities, and there must be an adequate stimulation that properly addresses the lags presented in each individual [58]. Differences in the development of gross and fine motor skills appear in DS children in the first months of life, with more active children achieving more marked development, highlighting the adaptability of early childhood and the importance of early stimulation [68], which also contributes to the development of functional skills, where people with DS underperform others [69].

### 4.2. Development of DS Communication

Language and communication in DS suffer the consequences of severe deficits in auditory processing and problems in visual skills that occur due to the characteristics of the syndrome phenotype [25]. In addition, in DS cases, language development is affected by morphological and anatomical aspects, which influence brain development, and abnormalities in the development of the cortex and cerebellum are probably substrates for neurocognitive impairment [17]. In addition, the non-acquisition of more complex means of communication and the delay in the acquisition and memorization of the language of action sequences, and changes in language processing that compromise understanding and social exchange [32], can lead to a cognitive delay that directly interferes in the development of the child with DS.

Language processing demands large structural brain connectivity, and changes in the brain generate changes in neural networks that can cause changes in stimuli and signals that pass from one neuron to another through synapses, as the brain of the child with DS has a deficit in the connections between neurons [17,18,28]. This may occur due to the lower total brain volume (TBV), lower gray matter in the frontal lobe and lower white matter, lower lobar cortical volume and lower hippocampal volume [70]. Regarding the reduction in TBV, specific sub-regions of the frontal lobe, temporal lobe, cerebellum and hippocampus can be highlighted [70], and these reductions will influence emotions, memory, balance, motor development and, consequently, language and cognition. These changes will also influence auditory processing, which causes difficulties in the recent auditory memory, which is the archived memory used to process, maintain, assimilate and understand the spoken language, in addition to controlling reaction times when it comes to interpreting or responding to language. In this way, the response skills in terms of time and intensity among children with DS are deficient, requiring a different method to stimulate these aspects [71]. In addition, research conducted with the use of computers found that individuals with DS have difficulty in transferring the task presented on the computer to a similar real situation [58].

Studies have pointed to the importance of using figures and visual clues, which lead to an increase in communicative initiatives in children with DS [61], in addition to pointing to the better understanding of the child with DS in tasks relating to the visual synthesis [72] of gestural communication, understood as the use of non-verbal gestures before the word appears, which is a strength in children with DS [23]. However, the psycholinguistic pattern in DS is not homogeneous in relation to auditory and visual processing. The profile of specific deficits suggests that educational support may need to be specific [25], but, from the perspective of this research, it is clear that the studies carried out so far show that electronic games for the evaluation and stimulation of children with DS can be a significantly important resource, although still little is known, considering the fact that there are no resources specifically developed for children with DS.

### 4.3. Use of Electronic Games for DS

Below is a table with the results of the integrative literature review carried out for this study and its analysis. In this research, 18 articles were found that met the inclusion criteria, published between the years 2011 and 2020, a number that can be considered to be low considering the benefits that virtual and computational technologies can offer for child development in general and, in particular, for children with DS, as shown by the results and conclusions of these articles. Although two of these articles do not refer to studies with children with DS, they were related in view of presenting perspectives of studies for such children and for bringing important information for the development of other research in the area. Of these studies, only one used a game created specifically for children with DS. Among the studies found, nine were carried out in Brazil, two in the United Kingdom, two in Spain, one in Chile, one in Taiwan, one in the United States and one in Hungary (Table 4).

In article n^o^ 11, described in Table 4, there is a history of computational technology pointing to its arrival in education in the 1970s in a process of social, scientific and technological evolution, recently leading to the use of the internet and applications for the stimulation of children’s development and learning [57]. In that study, it is emphasized that a virtual environment can facilitate the learning of a child, especially children with DS, showing itself to be a natural way of exploring the realities of day-to-day life, a fact corroborated in a study carried out using an electronic game called “Papado” (n^o^ 12), where it was verified to be an important contribution of technology regarding the learning of symmetry, colors, figures, ordinality, set, quantity, addition and subtraction for children with DS [40].

Still regarding the learning of mathematical concepts, a virtual space was used to seek the development of counting skills for young people and adults (n^o^ 04), promoting entertainment and stimulating imagination and motivation within a virtual community [51]. This study does not talk about DS children; however, given the results and the means used by the researchers, we consider it important to mention the work in this study, as it verifies that the study must be replicated in children with DS. A technological tool based on Serious Games (n^o^ 08) aimed at people with intellectual disabilities, including DS, was also used, reaching the conclusion that it is possible to develop technological solutions that work successfully as a pleasant training tool and with functions of tele monitoring, with the aim of stimulating this group [54].

Another important tool among electronic games is virtual reality, used to check the psychomotor needs of a child with DS. The researchers used the Motor Development Scale and an Xbox 360 video game with the Kinect sensor (n^o^ 09) and, after 20 intervention sessions, they observed improvements in global motor skills, balance, body layout and spatial organization. However, the development of fine motor skills, language and temporal organization remained stable. The authors concluded that, since these skills are fundamental for the development of communication and reading and writing, it can be inferred that there will be an improvement in these aspects after using the proposed game [55]. In another study using the Kinect sensor, the game TANGO: H (n^o^ 16) was used for the visual–motor stimulation of children with DS, and the results of the observations showed an improvement in the visual–motor cognitive skills of the participants [39].

In research that also used virtual reality in children with DS (n^o^ 01), an improvement in their postural control was observed, pointing to the possibility of an improvement in motor skills with the continued use of the proposed activities [73]. Similarly, Wuang et al. [62] concluded that, after training in virtual reality with Nintendo^®^ Wii^TM^ games (n^o^ 17), there was an improvement in sensorimotor functions and, consequently, in motor, visual-integrative skills and in the functioning of sensory integration in children with DS. The authors highlighted that virtual reality demonstrated promising effects both in individuals with cerebral palsy and DS through sensorimotor activation, helping with functional activities. The authors also observed the importance of virtual reality as a daily practice in the acquisition and mastery of skills which have a basis between neural plasticity and the principles of motor learning, so that games provide visual and auditory biofeedback, possibly promoting the improvement of the motor skills and postural control necessary to successfully use such games [50]. In another study in which the Nintendo^®^ Wii ™ video game (n^o^ 05) was used, the researchers realized that the accelerometer is a good tool with which to assess the movement acceleration characteristics of children and adolescents with DS during virtual bowling and golf games, which can contribute to the maintenance of motor skills already acquired and stimulate new possibilities for movement [52].

Studies also point out that virtual reality (n^o^ 15) has demonstrated effectiveness in the acquisition and development of autonomous life activities, language skills, social skills and educational aspects of people with disabilities [61]. In addition, there is a benefit to using technology and video games, such as Nintendo Wii, in the development of motor skills in children and adolescents with atypical development. For Álvarez et al. [48], the use of this technology could benefit motor development, with promising effects on balance. However, he points out that more research is needed to prove its reliability and reproducibility.

Virtual reality was also used through a game called MoviPensando (n^o^ 07), where the child’s silhouette is transferred to the screen and they have to virtually touch objects that have indirect associations with each other, based on varied criteria, such as colors, equal quantities, complementary images (man–woman), pairs (drawing–name), etc. The objective of the game is the motor and cognitive stimulation of the child with DS [53].

Regarding fine motor skills, working with children aged 6 to 36 months, researchers used the touchscreen scroll (n^o^ 02), managing to find significant correlations between the use of the screen and the main landmarks of motor and language development in children, despite using only one finger to scroll the screen [49]. This fact leads us to infer that games that use more fingers can lead to the better development of fine motor skills. However, it should be noted that this study was not conducted with children with DS.

A virtual labyrinth task using a cell phone (n^o^ 10) was also used to verify the development over time of acquisition and retention skills in children with DS in aspects related to motor learning, realizing that, despite performing the tasks, the children with DS need more time than the other children to complete the maze [58]. Platforms were also tested in a study comparing six interactive games from Leap Motion, Nintendo platforms Wii^®^ and Timocco, investigating which aspects of the games were of more benefit, with aspects such as easiness and fun being the most relevant for DS children [59]. The computer virtual maze task was used in another study (n^o^ 13), where the researchers found that individuals with DS had ease of acquiring and retaining information, but had difficulty transferring the task presented on the computer to a real situation [58]. Likewise, in yet another study that proposed virtual route games (n^o^ 14), the authors realized that a weak nonverbal capacity related to DS can be detrimental to some aspects of route learning in the group that has Down syndrome [60].

The games listed above used clues and visual elements as the main element for the stimulation of children with DS, demonstrating the greater ease presented by individuals with DS in relation to vision. However, it is important to pay attention to the control of eye movement during the visualization of scenes, a very important fact for the development of a game that seeks to stimulate the development of children with DS, since, despite several studies on the subject, there is relatively small knowledge about the mechanisms that control the duration of a child’s gaze on the screen [74]. In this sense, Smith and Mital [75], in a study carried out on children without DS, noticed that dynamic scenes and areas of high flicker show greater attentional synchrony, longer fixations and greater attention on the part of people than static scenes. From this perspective, in a proposal for a photomontage and emotion recognition task, also performed on children without DS, it was noticed that exaggerated emotions facilitated the accuracy and speed of recognition by children, giving clues for the construction of games aimed at the development of the understanding of emotions [76] in those who lack these abilities.

Finally, research using the virtual environment “Our Life” (n^o^ 06), developed to help children with Down syndrome to memorize action sequences of their daily routine, concluded that the playful activities developed promoted the interest of children who, in addition to having fun, were able to test hypotheses and question their own sequence of actions performed daily [32], although the authors emphasize the need for the participation of teachers in the use of technologies for their success in the development of the DS children.

The games used in these studies were able to stimulate, through the visual field, the skills of global motor skills, balance, body scheme and spatial organization, in addition to the learning of mathematical concepts, in order to directly influence the autonomous life activities, language skills, social skills and educational aspects of people with DS. However, only one game was found to be specifically developed for this group, which makes it difficult to further analyze the results found. However, in view of the study of comorbidities presented in DS, it can be said that, first, it is necessary to be aware that for the development of electronic games for the stimulation of children with DS, it is necessary to observe their difficulties in relation to hearing, vision, ligament laxity, hypotonia, psychomotor development and cognitive impairment, among others, which can interfere with their performance and even access to such games. Thus, it must be considered that games need to have a possibility of adjustment or calibration that individualizes them for each child in order to provide conditions for everyone, regardless of the problems presented, to obtain satisfactory results without being harmed by health problems that may arise.

It was noticed that, although these studies present important perspectives for the stimulation of children with DS using virtual and computational technologies, there is still a gap in the literature regarding studies on the use of games developed specifically for that population, and only one was found in the research.

## 5. Conclusions

Despite the contribution of technology in relation to the teaching–learning of DS children through electronic games presented in the studies described in this research, this very important tool is still underestimated and needs to be deeply studied. The area is still lacking more research and investment in games and applications that can specifically assess and stimulate the neuro, psychomotor and cognitive development of children with DS.

At the end of this research, it was realized that there are still few studies using virtual and computational technologies for children with DS and that, despite the relative success presented in them, there is still a long way to go to find tools that can present significant results to be used in other research and that address the needs of the person born with DS.

## Figures and Tables

**Figure 1 ijerph-19-02955-f001:**
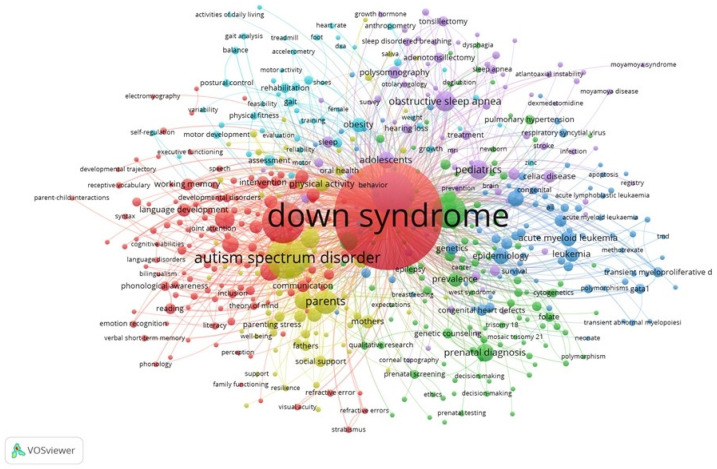
Co-occurrence network of keywords by the authors.

**Figure 2 ijerph-19-02955-f002:**
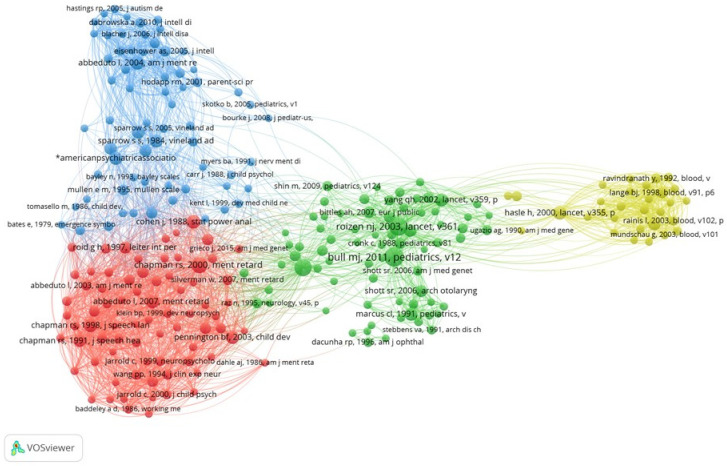
Citation network of cited references.

**Figure 3 ijerph-19-02955-f003:**
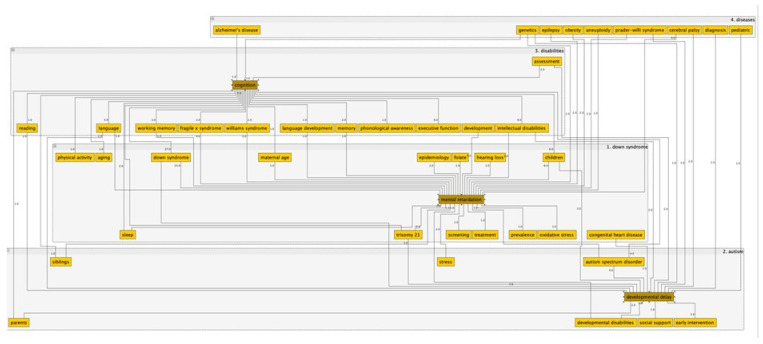
Conceptual space of the research, with the selection of the keywords “Down syndrome” and “games”.

**Figure 4 ijerph-19-02955-f004:**
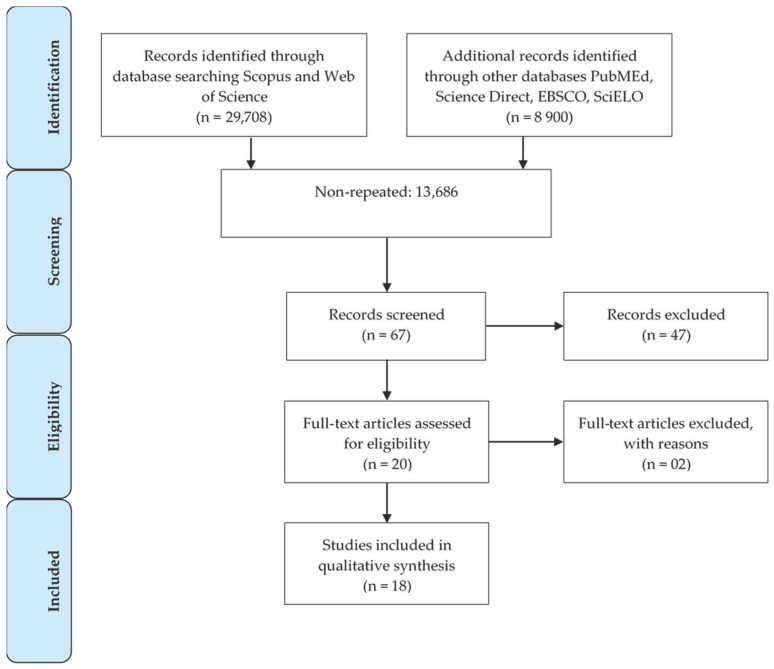
Stages of the search and selection process.

**Table 1 ijerph-19-02955-t001:** Summary of the main developmental characteristics of individuals with Down syndrome.

Development Area	Associated Issues	Main References in the Field
Health	Congenital heart diseases	Zhang, Liu, and Tian [9]; Parker et al. [3]; Roizen and Patterson [8]
	Thyroid dysfunction	Graber et al. [10]
	Obesity and early aging	Malegiannaki et al. [12]
Neurodevelopment	Premature aging of the brain and the immune system	Franceschi et al. [16]
	Abnormalities in the development and connectivity of the cortex and cerebellum	Patkee et al. [17]; Pennington et al. [18]
	Functional brain connectivity interrupted in the motor and prefrontal cortex	Xu et al. [19]
	Behavioral phenotype	Fidler [11]; Chapman and Hesketh [11]
Motor	Motor development delay	Kim et al. [20]
	Abnormal bone growth and ligamental connexon	Bertapelli et al. [6]
	Generalized muscle hypotonia and hyper-laxity	Kaczorowska et al. [2]; Antón et al. [21]; Mansour et al. [13]
	Problems with balance, coordination and manual co-ordination	Reis et al. [22]
Communication	Difficulty in expressive language	Linn et al. [23]; Martin et al. [24]
	Deficits in language and communication	López-Riobóo and Martínez-Castilla [25]; Roberts et al. [26]
Cognition	Intellectual disability	Rosser et al. [27]; Silverman [28]; Lanfranchi et al. [29]
Hearing	Hearing loss	Fisher [14]; Nightengale et al. [30]; Kreicher et al. [31]
Vision	Visual disorders	da Cruz Netto et al. [32] Tomita [33]

**Table 2 ijerph-19-02955-t002:** The most influential keywords in the co-occurrence network.

Keywords	Cluster	Average Year	Degree	Eigenvector Centrality
Down syndrome	1	2012.0341	546	1.0000
Children	1	2013.212	281	0.6343
Intellectual disabilities	3	2014.6408	265	0.6087
Autism spectrum disorder	4	2012.0855	215	0.5259
Trisomy 21	1	2012.5455	220	0.4839
Mental retardation	3	2005.6486	96	0.2901
Fragile X syndrome	3	2012.4706	90	0.2691
Parents	4	2013.7024	83	0.2601
Language	3	2010.619	77	0.2461
Developmental disabilities	4	2012.7241	71	0.2447
Cognition	3	2011.7317	68	0.2247
Williams syndrome	3	2012.0238	81	0.2224
Obstructive sleep apnea	1	2014.9101	67	0.2047
Prevalence	2	2011.7708	62	0.2029
Adolescents	6	2015.2105	54	0.2017
Development	3	2012.2439	60	0.2003
Autism spectrum disorders	4	2013.2955	54	0.1936
Genetics	1	2011.1176	60	0.1926
Siblings	4	2012.9783	47	0.1910
Epidemiology	2	2009.2105	60	0.1891
Quality of life	4	2015.8936	54	0.1883
Physical activity	6	2015.8696	53	0.1867
Sleep	1	2013.4194	49	0.1860
Obesity	6	2013.587	51	0.1835
Pediatric	1	2015.0816	69	0.1833
Developmental delay	3	2009.92	46	0.1787

**Table 3 ijerph-19-02955-t003:** Documents with the highest eigenvector centralities in the cited reference network.

Document	Subject	Degree	Eigenvector Centrality
Parker et al. [3]	The prevalence of birth defects	207	1.0000
Chapman and Hesketh [47]	Behavioral phenotype of individuals with Down syndrome	182	0.9913
Roizen and Patterson [8]	Medical management of Down syndrome	195	0.9001
Fidler [11]	The emerging Down syndrome behavioral phenotype in infants, toddlers and preschoolers	156	0.8846
Pennington et al. [18]	Neuropsychological domains of individuals with Down syndrome	153	0.8783
Silverman [28]	Cognitive characteristics of Down syndrome	155	0.8660
Abbeduto et al. [34]	The syndrome-specific features of the language phenotype	141	0.8394
Roberts, Price and Malkin [26]	The language and communication development of individuals with Down syndrome	141	0.8345
Martin et al. [24]	The language and literacy skills of individuals with Down syndrome	140	0.8233
Lanfranchi et al. [29]	Executive function (EF) in adolescents with Down syndrome	138	0.8167

**Table 4 ijerph-19-02955-t004:** Characteristics of the selected studies.

n^o^	Authors	Year	Country	Technology	Main Findings
01	Álvarez et al. [48]	2018	Chile	Nintendo^®^ Wii™ TV	The intervention based on virtual reality was effective for the Wii game, since it provides low-impact exercises to improve postural control and, thus, leads to better performance in TGMD 2 in children with DS.
02	Bedford et al. [49]	2016	United Kingdom	Touchscreen technologies	In the present study, no evidence was found to support a negative association between the age of first use of the touchscreen and developmental milestones. Previous use of the touchscreen, specifically scrolling, was associated with previous fine motor performance. Future longitudinal studies are needed to elucidate the temporal order and the mechanisms of this association and to examine the impact of using the touchscreen on other more refined measures of behavioral, cognitive and neural development.
03	Berg et al. [50]	2012	USA	Nintendo^®^ Wii™ TV	The practice of the Wii game in children with DS can be a good tool for improvements in the coordination of the upper limbs, manual dexterity, balance, postural stability and control of stability limits.
04	Boleracki et al. [51]	2015	Hungary	Virtual Game	The study suggests that virtual space for DS will not only help in the development of counting skills for young adults, but will also create an entertainment environment for all visitors, in addition to promoting imagination and motivation within a virtual community.
05	Carrogi-Vianna et al. [52]	2017	Brazil	Nintendo^®^ Wii™ TV	The accelerometer is a good tool for assessing the movement acceleration characteristics of adolescents with DS during virtual bowling and golf played on the Nintendo^®^ Wii ™ video game.
06	da Cruz Netto et al. [32]	2020	Brazil	Virtual environment “Our Life”	According to specialists (psychologist and pedagogue) from APAE and parents, the recreational activities implemented in this virtual environment have been of great interest to children, who had fun, tested hypotheses and questioned them about the sequences of actions carried out in their daily lives.
07	Diatel et al. [53]	2016	Brazil	MoviPensando	MoviPensando is a Serious Game that uses a webcam to capture the child’s image and insert it into the game where “groups of similar images” or “from the same context” are used, thus expanding the cognitive scope achieved. The combination of cognitive and motor aspects in a fun activity makes MoviPensando an interesting and viable option for use by health and education professionals.
08	Lopez-Basterretxea et al. [54]	2014	Spain	Serious Games	Based on a technological tool based on Serious Games aimed at people with ID, including DS, it was concluded that it is possible to develop technological solutions that work successfully as a pleasant training tool and with remote monitoring functions.
09	Lorenzo et al. [55]	2015	Brazil	Xbox 360 with Kinect TV sensor	After the intervention, there was an improvement in the skills of global motor skills, balance, body scheme and spatial organization; however, the development of fine motor skills and language/temporal organization remained stable.
10	Menezes et al. [56]	2015	Brazil	Marble Maze Classic^®^—Mobile	It was concluded that there was a motor learning process in individuals with DS through the maze task on mobile devices, who showed improved performance, evidenced by a reduced time in the retention and maintenance phase in the transfer phase.
11	da Cruz Netto et al. [57]	2014	Brazil	Virtual environment “Our Life”	The proposed virtual environment provides entertainment, attractiveness and immersion for the child in the plot, which helps them to memorize the daily routines implemented.
12	de Oliveira et al. [40]	2017	Brazil	Papado (video game)	The use of the Papado electronic game, together with the mediation of a professional, provided children with DS with a learning process for symmetry, colors, figures, ordinality, set, quantity, addition and subtraction.
13	Possebom et al. [58]	2016	Brazil	Maze task—PC	It was found that participants with DS improved their performance during acquisition and retention, but showed difficulty in transferring the computational task to a similar situation.
14	Pelosi et al. [59]	2019	Brazil	Six interactive games from Leap Motion, Nintendo platforms Wii^®^ and Timocco	The correlation between the “players’ performance” and the “demonstration of interest” variables presented significant results. Game preference was based on how much fun and easy to play the game was when improving children’s performance.
15	Purser et al. [60]	2015	United Kingdom	Virtual Route Games	The study showed that weak non-verbal ability can be particularly detrimental to some aspects of route learning in the DS group.
16	Rodrigues et al. [61]	2015	Brazil	Video Modeling—PC	The results show that Video Modeling can be an effective and fast technique to teach the communication system by exchanging figures. The effects of this intervention in relation to this child with DS contributed to the increase in vocabulary in general, in addition to developing communication skills.
17	Torres-Carrión et al. [39]	2019	Spain	TANGO: H Designer—Kinect sensor	Visual–motor cognitive stimulation through the movement of hands, arms, feet and head has been proposed. The TANGO: H platform allowed users to design and implement digital exercises for gestural interaction, adding to its set of uses the stimulation of cognitive visual–motor skills in individuals with DS. The gestural platform allows users to stimulate visual–spatial memory in individuals with DS, producing excellent results in all cases.
18	Wuang et al. [62]	2011	Taiwan	Nintendo^®^ Wii ™—TV	Virtual reality using Wii gaming technology has demonstrated benefits in improving sensory motor functions among children with DS. It can be used as adjunctive therapy for other successful rehabilitation interventions proven to treat children with DS.

Note: Test of Gross Development (TGMD-2).

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
