# Peer review of "The Use of Virtual and Computational Technologies in the Psychomotor and Cognitive Development of Children with Down Syndrome: A Systematic Literature Review"

_ijerph, 2022, doi:10.3390/ijerph19052955_

Round 1

Reviewer 1 Report

According to authors, the aim of this article was to carry out an integrative review of the literature on the use of 13 virtual and computational technologies in the stimulation of children with Down syndrome (DS). A 14 search and analysis of articles published between 2011 and 2020 in the PubMed, Scopus, Science 15 Direct, EBSCO, SciELO and Web of Science databases was carried out.

The paper has two very small sections: 2 and 3. I recommend authors to join this section in one more consistent section.

Authors use the acronyms DS and SD to refer to the term "Down Syndrome" throughout the whole paper. It's necessary to uniformize the use of acronyms.

It would be interesting if authors provide their own view about this research area, how it can be evolved and how new and innovative proposals can be constructed in order to really contribute to the evolution in the state-of-the art according their own experience in this research area. This analysis can be written at the end of section 5 or 6. This kind of analysis is very interesting in literature review papers.

Author Response

Carla Cristina Vieira Lourenço

Address: Rua Coração de Jesus Bloco 4 1ºDt Frente 3510-774 Viseu

Phone: +351964890040

                                                                                                                                                          February 14th, 2022

To

Professor Cicely Chen

Editor
International Journal of Environmental Research and Public Health

Ref. Revised version of the manuscript ( IJERPH-1555201 ): “THE USE OF VIRTUAL AND COMPUTATIONAL TECHNOLOGIES IN THE PSYCHOMOTOR AND COGNITIVE DEVELOPMENT IN CHILDREN WITH DOWN SYNDROME: A SYSTEMATIC LITERATURE REVIEW”

Dear Editor,

Thank you for the opportunity to resubmit our manuscript. The comments from the reviewers were highly insightful and enabled us to improve the quality of our manuscript. We have changed it according to their advice. A point-by-point response to the reviewers, as well as the marked-up version (highlighted in red) of the manuscript, is attached.

I believe that we have addressed all questions raised by the reviewers, and the manuscript has greatly improved. Thank you again for your time and effort in considering this manuscript for publication.

Best regards,

Carla Lourenço

Response to Reviewer 1 Comments

 According to authors, the aim of this article was to carry out an integrative review of the literature on the use of 13 virtual and computational technologies in the stimulation of children with Down syndrome (DS). A 14 search and analysis of articles published between 2011 and 2020 in the PubMed, Scopus, Science 15 Direct, EBSCO, SciELO and Web of Science databases was carried out.

Dear reviewer, thank you for your valuable comments. All raised comments were adressed in the response letter. You will find our notes in “red” fonted text.

Comment 1: The paper has two very small sections: 2 and 3. I recommend authors to join this section in one more consistent section.

Response 1: Thank you, for your valuable comments. We modified as suggested by the review.

Comment 2:  Authors use the acronyms DS and SD to refer to the term "Down Syndrome" throughout the whole paper. It's necessary to uniformize the use of acronyms.

Response 2: Thank you, for your valuable comments. We modified as suggested by the review. Line 31 (and all subsequent uses)

Comment 3: It would be interesting if authors provide their own view about this research area, how it can be evolved and how new and innovative proposals can be constructed in order to really contribute to the evolution in the state-of-the art according their own experience in this research area. This analysis can be written at the end of section 5 or 6. This kind of analysis is very interesting in literature review papers.

Response 3: Thank you, for your valuable comments. During data analysis, the following steps were taken: To identify emerging themes. The co-occurrence analysis of keywords was used to identify trends in the focus of the topics studied about this research. Zupic and Cater (2015) stated that when words often co-occur in documents, it means that the concepts of those words are closely related. After that, it is necessary to run the co-occurrence network of keywords with only one occurrence. The metadata was imported by VOSviewer (VAN ECK, WALTMAN, 2010) and the authors' keyword co-occurrence network was generated[42]. Then, import into Gephi according to the Bastian et al. (2009) and sort by the column “score<avg._pub._year>”.

Reference: Bastian M, Heymann S, Jacomy M: Gephi: an open source software for exploring and manipulating networks. In: Proceedings of the international AAAI conference on web and social media: 2009; 2009: 361-362.

Van Eck N, Waltman L: Software survey: VOSviewer, a computer program for bibliometric mapping. scientometrics 2010, 84(2):523-538.

Zupic I, ÄŒater T: Bibliometric methods in management and organization. Organizational research methods 2015, 18(3):429-472.

Reviewer 2 Report

The article (Manuscript ID: ijerph-1555201) provides useful information on " The use of virtual and computational technologies in the psy-chomotor and cognitive development in children with Down syndrome: a systematic literature review ". First, thank you for allowing me to read this paper which initially deals with an interesting and relevant topic. The suggestions given in these points are intended to improve your work.

(1) Improve the introduction section, providing the problem

(2) In all text: arrange the references from the oldest to the newest in the text such as lines 25 " Holz et al. 2019; Kaczorowska et al. 2019; Parker et al. 2010", Line 38 "Bull and the Committee on Genetics 2011; Parker et al. 2010", Lines 42-43 ….. etc.

(3) All figures are not in the manuscript

(4) Write the reference within the text according to the Journal instructions.

(5) Line 40: Add "a" before " negative impact"

(6) Line 41: Change " see summary at" to " see the summary in"

(7) Line 44: This reference "Fisher 2020" has not discussed thyroid disorders

(8) Lines 44-45: one reference is enough "Chapman and Hesketh 2000; Fidler 2005"

(9) Line 58: Put "." after " 2013)" to finish the sentence

(10) In Table 1: this reference " Nightengale et al. 2017" is not found in list of references

(11) In Table 1: this reference " Kreicher, Weir, Nguyen, & Meyer, 2018" is not found in list of references

(12) In Table 1: this reference " Tomita, 2017" is not found in list of references

(13) Line 94: Change " a literature review article" to "literature review articles"

(14) Line 100: Add "the" before " title"

(15) Line 116: Change " about" to "in"

(16) Line 139: Table 1 or Table 2 is correct? Please check

(17) Line 170: Change "47 paper " to "47 papers"

(18) It is preferable to shorten the "6. Conclusions"

(19) Revise the references inside and outside the text

Author Response

Carla Cristina Vieira Lourenço

Address: Rua Coração de Jesus Bloco 4 1ºDt Frente 3510-774 Viseu

Phone: +351964890040

                                                                                                                                                          February 14th, 2022

To

Professor Cicely Chen

Editor
International Journal of Environmental Research and Public Health

Ref. Revised version of the manuscript ( IJERPH-1555201 ): “THE USE OF VIRTUAL AND COMPUTATIONAL TECHNOLOGIES IN THE PSYCHOMOTOR AND COGNITIVE DEVELOPMENT IN CHILDREN WITH DOWN SYNDROME: A SYSTEMATIC LITERATURE REVIEW”

Dear Editor,

Thank you for the opportunity to resubmit our manuscript. The comments from the reviewers were highly insightful and enabled us to improve the quality of our manuscript. We have changed it according to their advice. A point-by-point response to the reviewers, as well as the marked-up version (highlighted in red) of the manuscript, is attached.

I believe that we have addressed all questions raised by the reviewers, and the manuscript has greatly improved. Thank you again for your time and effort in considering this manuscript for publication.

Best regards,

Carla Lourenço

The article (Manuscript ID: ijerph-1555201) provides useful information on " The use of virtual and computational technologies in the psy-chomotor and cognitive development in children with Down syndrome: a systematic literature review ". First, thank you for allowing me to read this paper which initially deals with an interesting and relevant topic. The suggestions given in these points are intended to improve your work.

Dear reviewer, thank you for your valuable comments. All raised comments were adressed in the response letter. You will find our notes in “red” fonted text.

Comment 1 Improve the introduction section, providing the problem

Response 1 Thank you, for your valuable comments. We modified as suggested by the review. See section “Introduction”. Line 82 to 94.

Comment 2 In all text: arrange the references from the oldest to the newest in the text such as lines 25 " Holz et al. 2019; Kaczorowska et al. 2019; Parker et al. 2010", Line 38 "Bull and the Committee on Genetics 2011; Parker et al. 2010", Lines 42-43 ….. etc.

Response 2 Thank you, for your valuable comments. We modified as suggested by the review. See full article.

Comment 3 All figures are not in the manuscript

Response 3 We apologize for not adding a figure in correct places. We added all figures.

Comment 4 Write the reference within the text according to the Journal instructions.

Response 4 Thank you, for your valuable comments. We modified as suggested by the review. See section “References”

Comment 5 Line 40: Add "a" before " negative impact"

Response 5 We modified as suggested by the review. See line 46

Comment 6 Line 41: Change " see summary at" to " see the summary in"

Response 6 We modified as suggested by the review. See line 47

Comment 7  Line 44: This reference "Fisher 2020" has not discussed thyroid disorders

Response 7 We really appreciate the comments of the reviewer. We added a reference that aimed to discuss Down syndrome and thyroid disorders. Line 49.

Reference: Graber, E., Chacko, E., Regelmann, M. O., Costin, G., & Rapaport, R. (2012). Down syndrome and thyroid function. Endocrinology and Metabolism Clinics41(4), 735-745.

Comment 8 Lines 44-45: one reference is enough "Chapman and Hesketh 2000; Fidler 2005"

Response 8 We completely agree with your comment. We removed Fidler (2005) reference.

Comment 9 Line 58: Put "." after " 2013)" to finish the sentence

Response 9 We modified as suggested by the review.

Comment 10 In Table 1: this reference " Nightengale et al. 2017" is not found in list of references

Response 10 We apologize for not adding a full reference. We added as suggested by the reviewer.

Reference: Nightengale, E., Yoon, P., Wolter-Warmerdam, K., Daniels, D., & Hickey, F. (2017). Understanding hearing and hearing loss in children with Down syndrome. American Journal of Audiology26(3), 301-308.

Comment 11 Table 1: this reference " Kreicher, Weir, Nguyen, & Meyer, 2018" is not found in list of references

Response 11 We apologize for not adding a full reference. We added a reference.

Reference: Kreicher, K. L., Weir, F. W., Nguyen, S. A., & Meyer, T. A. (2018). Characteristics and progression of hearing loss in children with Down syndrome. The Journal of pediatrics193, 27-33.

Comment 12  In Table 1: this reference " Tomita, 2017" is not found in list of references

Response 12 We apologize for not adding a full reference. We added a reference.

Reference: Tomita, K. (2017). Visual characteristics of children with Down syndrome. Japanese journal of ophthalmology61(3), 271-279.

Comment 13  Line 94: Change " a literature review article" to "literature review articles"

Response 13 We modified as suggested by the review. See line 110

Comment 14  Line 100: Add "the" before " title"

Response 14 We modified as suggested by the review. See line 116.

Comment 15 Line 116: Change " about" to "in"

Response 15 We modified as suggested by the review.

Comment 16 Line 139: Table 1 or Table 2 is correct? Please check

Response 16 We apologize for confusing the correct order of the tables. The correct one is table 2. See line 151.

Comment 17 Line 170: Change "47 paper " to "47 papers"

Response 17 We modified as suggested by the review. See line 163.

Comment 18 It is preferable to shorten the "6. Conclusions"

Response 18 Thank you, for your valuable comments. We modified as suggested by the review. See section “Conclusion”

Comment 19 Revise the references inside and outside the text

Response 19 We completely agree with your comment. The reference section was reorganized.

Reviewer 3 Report

I believe the authors tap into an important topic in children with Down syndrome health.

This integrative review aims to analyse the literature on the use of virtual and computational technologies in the stimulation of children with Down syndrome (DS)). This paper tries to explain in a positive way when looking at the topic of health in this specific population. I think some things need clarifying for the publication that will help in the overall interpretation and understanding of the results before being published within the scope of IJERPH.

Abstract

Comment 1: The authors cited “…. for the articles occurred through the following descriptors in English and Portuguese”, but in 2. Literature Search (line 90) the authors wrote “Language was limited to English, Spanish, French and Portuguese “. I am confused, can the authors clarify this?

Comment 2: line 15 - For the readers, it is crucial to understand the main results of this review. Please rewrite the abstract according to IJERPH structure.

Eligibility criteria

Comment 3: (line 105) – “…selected was 18 searches (Table 4)”. To better understand the information, the authors should be to insert Table  4 following this paragraph.

Data collection

Comment 4: (line 145) “Table 1 shows the most influential keywords in the co-occurrence network”. Please change to Table 2…

Comment 5: (line 163) Figure 3 -  To better understand the picture, the authors should draw the new figure.

Comment 6: (line 167) The authors wrote “After the removal of redundancies 13.689….”. However, in  Figure 4   the 13.689 were classified as “non-repeated” Can the authors clarify this?

Comment 7: (line 274) “In the oldest article (No. 11) described in Table 1, there is a history of computational  technology pointing to its arrival at education in the 1970s in a process of social, scientific.” The authors referenced table 1, but this information is not available in table 1. Can the authors clarify this?

Conclusion

Comment 8: The conclusion should be more accurate and precise. I suggest no more than 1 or 2 paragraph (s).

General comment: The authors did hard and deep work, notwithstanding structural and other minor errors that difficulted the lecture and data interpretation. I suggest a deep reorganization of this important information, for better interpretation of the readers.

Author Response

Carla Cristina Vieira Lourenço

Address: Rua Coração de Jesus Bloco 4 1ºDt Frente 3510-774 Viseu

Phone: +351964890040

                                                                                                                                                          February 14th, 2022

To

Professor Cicely Chen

Editor
International Journal of Environmental Research and Public Health

Ref. Revised version of the manuscript ( IJERPH-1555201 ): “THE USE OF VIRTUAL AND COMPUTATIONAL TECHNOLOGIES IN THE PSYCHOMOTOR AND COGNITIVE DEVELOPMENT IN CHILDREN WITH DOWN SYNDROME: A SYSTEMATIC LITERATURE REVIEW”

Dear Editor,

Thank you for the opportunity to resubmit our manuscript. The comments from the reviewers were highly insightful and enabled us to improve the quality of our manuscript. We have changed it according to their advice. A point-by-point response to the reviewers, as well as the marked-up version (highlighted in red) of the manuscript, is attached.

I believe that we have addressed all questions raised by the reviewers, and the manuscript has greatly improved. Thank you again for your time and effort in considering this manuscript for publication.

Best regards,

Carla Lourenço

I believe the authors tap into an important topic in children with Down syndrome health.

This integrative review aims to analyse the literature on the use of virtual and computational technologies in the stimulation of children with Down syndrome (DS)). This paper tries to explain in a positive way when looking at the topic of health in this specific population. I think some things need clarifying for the publication that will help in the overall interpretation and understanding of the results before being published within the scope of IJERPH.

Dear reviewer, thank you for your valuable comments. All raised comments were adressed in the response letter. You will find our notes in “red” fonted text.

Abstract

Comment 1: The authors cited “…. for the articles occurred through the following descriptors in English and Portuguese”, but in 2. Literature Search (line 90) the authors wrote “Language was limited to English, Spanish, French and Portuguese “. I am confused, can the authors clarify this?

Response 1: We really appreciate the comments of the reviewer.  We made the necessary changes in the manuscript to better the reader’s understanding. We followed your prompts and marked all in red. Please refer to "Abstract" section, Page 1. Also, see in the section 2. Literature Search we removed this frase: “Language was limited to English, Spanish, French and Portuguese”.

I believe the authors tap into an important topic in children with Down syndrome health.

This integrative review aims to analyse the literature on the use of virtual and computational technologies in the stimulation of children with Down syndrome (DS)). This paper tries to explain in a positive way when looking at the topic of health in this specific population. I think some things need clarifying for the publication that will help in the overall interpretation and understanding of the results before being published within the scope of IJERPH.

Abstract

Comment 1: The authors cited “…. for the articles occurred through the following descriptors in English and Portuguese”, but in 2. Literature Search (line 90) the authors wrote “Language was limited to English, Spanish, French and Portuguese “. I am confused, can the authors clarify this?

Response 1: We really appreciate the comments of the reviewer.  We made the necessary changes in the manuscript to better the reader’s understanding. We followed your prompts and marked all in red. Please refer to "Abstract" section, Page 1. Also, see in the section 2. Literature Search (Line 82) we removed this frase: “Language was limited to English, Spanish, French and Portuguese”.

Comment 2: line 15 - For the readers, it is crucial to understand the main results of this review. Please rewrite the abstract according to IJERPH structure.

Response 2Thank you, for your valuable comments. We modified as suggested by the review. See section “Introduction” and Abstract.

Eligibility criteria

Comment 3: (line 105) – “…selected was 18 searches (Table 4)”. To better understand the information, the authors should be to insert Table 4 following this paragraph.

Response 3 We modified as suggested by the review. See line 174.

Data collection

Comment 4: (line 145) “Table 1 shows the most influential keywords in the co-occurrence network”. Please change to Table 2…

Response 4 We apologize for confusing the correct order of the tables. We make the adjustment on line 139 from “Table 1 shows the most influential keywords in the co-occurrence network.”to Table 2 shows the most influential keywords in the co-occurrence network.” The line 154 is correct.

Comment 5: (line 163) Figure 3 - To better understand the picture, the authors should draw the new figure.

Response 5 Unfortunately, the figure 3 is generated by the software yEd. yEd is a general-purpose diagramming program with a multi-document interface. There is no way to edit it

Comment 6: (line 167) The authors wrote “After the removal of redundancies 13.689….”. However, in  Figure 4   the 13.689 were classified as “non-repeated” Can the authors clarify this?

Response 6 Thank you for identifying this confusing. There was a misunderstanding between the research team and our English translator. Please refer to 3. Data collection, Line 177 to 180.

“After identification, 24,922 duplicate articles were removed. Remaining 13,689 articles that were exclusively related to infants or children with DS.When performing the screening through inclusion criteria, 67 studies were included with the descriptors: Visuomotor Development (22) OR Digital Games (10) OR Virtual Reality (35).”

Comment 7: (line 274) “In the oldest article (No. 11) described in Table 1, there is a history of computational  technology pointing to its arrival at education in the 1970s in a process of social, scientific.” The authors referenced table 1, but this information is not available in table 1. Can the authors clarify this?

Response 7 We apologize for confusing the table number. The correct is Table 4. Also, we removed "oldest" before “article”. See line 281.

“... in the article (no. 11) described in Table 4”

Conclusion

Comment 8: The conclusion should be more accurate and precise. I suggest no more than 1 or 2 paragraph (s).

Response 8 Thank you, for your valuable comments. We modified as suggested by the review. See section “Conclusion”.

Round 2

Reviewer 3 Report

The authors did respond to all my questions.

Minor reviews:

Line 412 - Please add an abbreviation DS (Down syndrome).

Line 415 - Please add an abbreviation DS (Down syndrome) 

Author Response

Dear Editor,

Thank you for the opportunity to resubmit our manuscript. The comments from the reviewer were highly insightful and enabled us to improve the quality of our manuscript. A point-by-point response to the reviewer, as well as the marked-up version (highlighted in red) of the manuscript, is attached.

Best regards,

Carla Lourenço

Response to Reviewer 3 Comments

Minor reviews:

Line 412 - Please add an abbreviation DS (Down syndrome).

Line 415 - Please add an abbreviation DS (Down syndrome) 

Response: Thank you, for your valuable comments. We modified as suggested by the review. Line 412 and line 415. You will find our notes in “red” fonted text.